# Factors perceived to facilitate or hinder handwashing among primary students: a qualitative assessment of the Mikono Safi intervention schools in NW Tanzania

Elialilia Okello [1,2] Saidi Kapiga,[1,2,3] Heiner Grosskurth,[1,2,3] Kenneth Makata,[1,2] Onike Mcharo,[1,2] Safari Kinungh'i,[2] Robert Dreibelbis[4]

[1]Mwanza Intervention Trials Unit, Mwanza, Tanzania
[2]National Institute for Medical Research (NIMR), Mwanza Centre, Mwanza, Tanzania
[3]Department of Infectious Disease Epidemiology, London School of Hygiene and Tropical Medicine, London, UK
[4]Department of Disease Control, London School of Hygiene and Tropical Medicine (LSHTM), London, UK

**Correspondence to**
Dr Elialilia Okello;
elialilia.okello@mitu.or.tz

## ABSTRACT

**Objective** To qualitatively assess the effects of a multi-modal school-based water, sanitation and hygiene (WASH) intervention on handwashing behaviour among primary students in North Western (NW) Tanzania.

**Design** The study was a qualitative assessment of barriers and facilitators to handwashing among students attending primary schools participating in the Mikono Safi Trial (Kiswahili for 'Clean Hands'), a cluster-randomised trial assessing the impact of a school-based WASH intervention on selected soil transmitted helminth infections. Data collection methods included in-depth interviews with teachers, focus group discussions and friendship pair interviews with students collected between April and October 2018. The Capability-Opportunity-Motivation and Behaviour model was used to inform data collection and analysis.

**Setting** The study was conducted in four purposively selected intervention schools in three districts of Kagera region, NW Tanzania (Bukoba urban, Bukoba rural and Muleba districts).

**Participants** Participants comprised 16 purposively selected teachers aged between 23 and 52 years and 100 students aged 7–15 years

**Results** The Mikono Safi intervention increased students' reported capability and motivation to wash their hands with soap at key times, particularly after visiting the toilet. Improvements in students' handwashing knowledge and skills were reported by both teachers and students, and motivation for handwashing was enhanced by emotional drivers such as disgust, fear and nurture. Newly established handwashing stations improved the physical opportunity to wash hands, although the availability of water and the provision of soap was not always consistent (eg, due to internal organisational shortcomings or during the dry season). Students and teachers were actively engaged in intervention implementation which created a school community that valued and supported improved hand hygiene.

**Conclusion** The intervention was successful in improving capability and motivation for handwashing. Handwashing opportunity was also greatly improved, although the supply with water and soap was sometimes interrupted, calling

## Strengths and limitations of this study

- ► In-depth interviews of teachers and focus group discussions and in-depth friendship pair interviews proved to be suitable methods to examine barriers to and facilitators of appropriate handwashing behaviour.
- ► These methods generated consistent results suggesting that the Mikono Safi intervention was effective in improving handwashing behaviour
- ► This was a qualitative study involving a small purposively selected sample of schools, teachers and pupils and may therefore not be representative for all eight intervention schools participating in the trial or for other primary schools in Tanzania.

for much stronger multi-sectoral collaboration to improve access to water at schools.

**Trial registration number** ISRCTN45013173; Pre-results.

## INTRODUCTION

Soil-transmitted helminth (STH) infections are among the most prevalent infections worldwide, affecting over 1.5 billion people.[1] These infections have demonstrable negative impacts on child development.[2 3] Current control strategies focus on school-based mass drug administration (MDA) programmes. Since 2006 the WHO recommends annual deworming among all at-risk people living in endemic areas if the population prevalence of soil-transmitted helminth infections exceeds 20%, and twice a year if it exceeds 50%, using orally prescribed albendazole (400 mg) or mebendazole (500 mg) which are both deemed effective, inexpensive and easy to administer by non-medical personnel (eg, teachers).[4] In 2009, the Tanzanian government began school-based MDA for STH control. Kagera, a region situated in North

Western Tanzania west of Lake Victoria, has been a major focus for this MDA initiative.[5] School children of this area have received annual treatment with albendazole for the past 8 years. However, recent survey data on the distribution of STH in Tanzania shows that the prevalence of these infections has remained high,[6] which is in keeping with recent research showing rapid re-infection following MDA programmes when water, sanitation and hygiene (WASH) conditions remain poor.[7] A recent randomised controlled trial in China demonstrated that the promotion of handwashing with soap in schools can result in significant reductions in STH prevalence,[8] suggesting that school-based WASH interventions could play a significant role in STH infection control.

The Mikono Safi (Kiswahili for 'clean hands') intervention was designed to increase handwashing with soap (HWWS) among school-aged children in an attempt to sustain the effects of routine MDA. It is a multi-modal intervention, combining health education, games and stories targeting key motivational drivers, improvements in school handwashing infrastructure and training and support to school staff. Training for teachers in all the intervention school was done by a qualified public health specialist and a social scientist with a Master's degree in sociology. Further details are provided below. The impact of the Mikono Safi intervention is being assessed in an ongoing cluster-randomised trial. As part of the trial, the STH prevalence in eight randomly selected schools in the Kagera region receiving the Mikono Safi intervention will be compared with eight schools in a randomly selected control arm.

While school-based WASH interventions in low-income and middle-income countries have been examined with regards to a wide range of health and educational outcomes, including: diarrhoea, respiratory infection, school attendance and health outcomes in the domestic environment,[9–13] these studies have shown that the impact on such outcomes is inconsistent. For the most part, this variability has been viewed through the lens of *intervention fidelity* and *compliance*—the extent to which schools and/ or implementing organisations have provided WASH services in schools routinely and in a manner consistent with intervention protocols.[14 15] However, for school-based hygiene interventions to result in health or educational impacts, they must also successfully change students' hygiene behaviours. Focussing on the implementation and delivery of school-based interventions alone has the potential to ignore the more complex relationships between an intervention and the proximal determinants of individual pupil HWWS behaviours. While studies have documented or tested the extent to which school-based interventions have resulted in changes in specific hygiene behaviours,[16] more information is needed on the mechanisms through which interventions change behaviours.

The goal of our study was to explore the mechanisms by which and the extent to which the Mikono Safi intervention influenced the determinants of children's handwashing behaviour. Our primary research question was: to what extent has the Mikono Safi intervention influenced the capability, opportunity and motivation for HWWS among pupils attending intervention schools? Findings from this research will be used to modify current intervention approaches, to further understand the extent to which school-based interventions influence individual drivers of behaviour, and to provide contextual information for interpreting forthcoming trial results.

## THE MIKONO SAFI INTERVENTION

The Mikono Safi intervention was informed by the Behaviour Centred Design process[17] and combined a number of evidence-based components to foster and trigger handwashing behaviours at schools. These comprised: (i) class-room based teacher-led health education on the negative health effects and the transmission route of faeco-orally transmitted STHs, the importance of handwashing with water and soap at key times[18 19]; (ii) messaging and activities designed to trigger emotional drivers of HWWS such as disgust, fear and nurture with regards to hygiene; (iii) demonstration of correct handwashing procedures; (iv) parental engagement through meetings at which parents were individually informed of their children's infection status; (v) provision of improved handwashing stations with water and soap near school toilets; and (vi) application of environmental nudges to subconsciously trigger hygiene behaviour.[16 20] The latter consisted of marked paths connecting toilets with handwash stations and painted hands on the hand-wash devices (figure 1). Health education in class comprised three sessions of about 2 hours each, making use of games, posters and sets of comic-type pictures of two students:

**Figure 1** Handwashing facility with 'nudges' installed in the intervention schools.

Koku, a young girl who both washed hands at key times and supported others to adopt good hygiene behaviour, and Muta, a young boy who learnt about appropriate hygiene behaviour in the process. Prior to implementation in each school, the study team completed a half-day training at each participating study school with eight to 10 teachers to train them on education and messaging activities, determine the schedule for school-based activities, and develop school-specific maintenance plans for handwashing facilities and materials. Full details on the Mikono Safi intervention are described in a forthcoming publication (Makata *et al*, in review)

## METHODS
### Theoretical framework
There are multiple models and frameworks that have been proposed to understand WASH behaviours.[17 21–23] For our study we chose the Capability-Opportunity-Motivation and Behaviour (COM-B) framework[24] to inform study design, data collection and analysis. COM-B, relative to other behaviour change theories and models, provides an intuitive and flexible approach to understanding behaviours that is applicable across a number of behaviours and contexts. COM-B identifies three broad categories of determinants of behavioural outcomes: capability, opportunity and motivation. Our study was focussed on HWWS. As such, we refer to capability as an individual's psychological and physical capacity to complete HWWS. Key aspects of capability include: handwashing skills, knowledge related to HWWS and self-efficacy. We define opportunity as consistent access to soap and water for handwashing at a location that is convenient for children and proper social and education support. Motivation is defined as the mental processes that direct behaviour. For the purposes of our study, we include cognitive, rational thought processes related to HWWS; habits; and emotional or motivational responses as part of individual motivation.

### Study design, sample and data collection methods
This was a qualitative study where we completed a range of qualitative data collection activities with students and teachers at four schools receiving the Mikono Safi intervention over a 6-month period. We used purposive sampling to select schools reflecting the range of schools participating in the Mikono Safi Trial. This included selected two urban and two rural schools for participation in our qualitative assessment from the four urban and four rural schools participating in the Mikono Safi Trial. We purposively sampled schools with both high and low pretrial STH infection prevalence. At the school-level, selection of teachers ensured inclusion of school administration, teachers in-charge of sanitation and other teachers who were involved in implementing hand hygiene education. Selection of students ensured inclusion of boys and girls, both younger and older children.

In each school, we completed two rounds of data collection. The first round of data was collected in April

| Table 1 | Summary of data collection activities |
|---|---|
| **February 2018** | **Intervention implementation** |
| April 2018 | Data collection, Round 1<br>► Two focus group discussions completed at each participating school<br>► Four in-depth interviews with teachers completed at each participating school |
| October 2018 | Data collection, Round 2<br>► Five friendship pair interviews completed at each participating school<br>► Four in-depth interviews with teachers completed at each participating school |

2018; 2 months after the Mikono Safi intervention had been launched in the four schools. The second round was collected in the last week of October 2018; (table 1). The purpose of conducting two rounds of data collection 6 months apart was to assess if the reported changes in behaviour and identified barriers and facilitators to proper HWWS changed over time.

In the first round of data collection, we completed two focus group discussions (FGDs) in each school, resulting in a total of eight FGDs with a total of 60 students. The number of FGD participants ranged between seven and 10. The FGDs were age and gender segregated—resulting in two FGDs each for older boys, older girls, younger boys and younger girls. During the FGDs, we explored students' shared perceptions about the influence of the intervention on handwashing. We also assessed students' handwashing skills: at the end of each FGD, the field researcher walked with the participating students to the handwashing stations for students to demonstrate how they normally washed hands. Handwashing education sessions had a practical component that took students through five handwashing steps which included: (i) wetting hands, (ii) putting soaps, (iii) scrubbing the palms together, (iv) scrubbing between the fingers and nails while counting up to 10 and (v) rinsing hands with running water. Handwashing skills for each student were assessed during observation by noting how many of the five recommended steps of handwashing were completed.

In addition to FGDs, we conducted four in-depth interviews (IDIs) with teachers to collect information on their perceptions about students' knowledge regarding the relationship between HWWS and STH infection; experience with implementing handwashing lessons; availability and accessibility of handwashing materials at school, and perceived effectiveness of the intervention in targeting motivational and emotional drivers for handwashing.

For the second round of data collection (October 2018), we completed five friendship pair interviews at each school, for a total of 10 respondents per school. FPI is a method used to collect data from children where the target respondent is encouraged to come to the interview with a friend in order to create a conducive, non-threatening environment for the children.[25] The research team, with support from respective class teachers,

solicited student volunteers. In each class, one student was selected. The selected student was asked to request his/her best friend to join the interview. Both the FGDs and FPIs collected information on students' knowledge regarding the relationship between HWW and STH infection, perceptions about implementation of handwashing lessons (including their perception about how consistent was the delivery of the handwashing messages), availability and accessibility of handwashing materials at school, and connections between education materials and targeted motivational and emotional drivers of HWWS. Whereas FGDs were used to gain insights into children's shared understanding of how the intervention components had influenced handwashing and the ways in which individuals influenced by others in a group situation, we used FPIs to gain insight into individual children's practices and personal experiences. During the second round of data collection, we also completed semi-structured interviews with four teachers per school, these were the same teachers interviewed in round one.

Data was collected using pilot tested interview guide (online supplementary file) . All interviews and FGDs were conducted in private empty rooms provided by the school administration. Each FGD/FPI lasted between 1 hour 30 min and 2 hours while interviews with teachers lasted between 30 and 45 min. All the interviews were collected in Kiswahili, transcribed, and translated in English by a bilingual speaker for analysis.

### Research team

The data collection team comprised three Tanzanian researchers (two social scientists and 1 medical doctor) at PhD or MSc level (ESO; KM, OM) and two research assistants, one male and one female with experience in qualitative research methods. Prior to starting fieldwork the research assistants received additional 1 week's training in qualitative research methods. The first author (ESO), participated in actual data collection at the beginning of each data collection round and coordinated the study. Before each round of data collection the study team visited the schools, explained the purpose of the study, discussed study activities with the school administration and made appointments for interviews and FGDs.

### Public involvement

This qualitative study is assessing an intervention that was preceded by rigorous formative research. During the formative research there was extensive involvement of students, teachers and regional and district education officers and school WASH coordinators in developing, testing and modifying the various components of the intervention. The feedback received from these stakeholders informed the design and implementation of the intervention. Similarly, the interview and FGD guides used in the qualitative study were pilot-tested and revised to include feedback from the participants. After the end of the research, the findings from the main trial and from this qualitative study will be disseminated through stakeholders meetings.

### Data analysis

Audio recorded interviews and FGDs were transcribed in Kiswahili and translated into English by the lead author and combined with field notes ready for analysis. Data were analysed using Atlas.ti V.7, a qualitative data analysis and research software

The COM-B conceptual framework served as a guide for data analysis and was used to define themes in analysis. A thematic analysis approach was used to facilitate the structuring, description and interpretation of results.[26] During analysis, the first author read all interview and FGD transcripts and formulated draft codes based on recurring accounts and descriptions that were identified in the transcripts. The first and last authors reviewed, discussed and refined these initial codes and then mapped each code back to the COM-B framework. There were no significant disagreements about how specific codes mapped back to the COM-B Framework. However, there were discussions about codes that conflated one or more COM-B determinants. These codes were then revised into subcodes that better mapped to an individual COM-B determinant. Each code was further classified into facilitators of and barriers to HWWS. After the coding structure was refined, the first author applied the codes to all interview and FGD transcripts and prepared summary memos on specific groups of related codes.

Data from both rounds were analysed together using the same methods. Data from each of the two rounds of data collection were then directly compared.[26]

### Ethical considerations

All participants provided informed assent (students) and consent (teachers); and parental consent was obtained for all participating students. Permission to interview students was sought from parents/guardians, while permission to interview teachers and school staff was sought from the school authorities. Personal data was anonymised, using ID numbers and stored data was stripped of names, and password protected for use by named research staff only.

### RESULTS

A total of 60 students participated in eight FGDs and another group of 40 students participated in 20 FPIs. On average a student's FGD/FPI lasted for 1 hour 40 min while teachers interview was about 40 min. In addition, IDIs were held with 16 teachers who were interviewed 2 months after the start of intervention and again 8 months after intervention had been rolled out. (table 2). All teachers and students selected for participation provided consent/assent and took part in the study.

Factors that study participants identified as facilitators of good handwashing behaviour were classified as according to the COM-B conceptual framework. Factors related to *capability* comprised knowledge and understanding of

**Table 2** Gender and age range of study participants

| | Study participants | | | | | |
| | Students | | | Teachers | | |
| Data collection round | Male | Female | Age range | Male | Female | Age range |
| --- | --- | --- | --- | --- | --- | --- |
| Round 1 | 30 | 30 | 7–15 years | 7 | 9 | 23–52 years |
| Round 2 | 20 | 20 | 8–14 years | 7 | 9 | 23–52 years |

the relationship between handwashing and STH, knowledge of correct handwashing; practical skills regarding handwashing following practical demonstrations of handwashing. Factors related to *social and physical opportunity* included the successful development of a school community of students and teachers who supported and encouraged handwashing; and the improved quality, number and location of handwashing stations with access to both water and soap. Reflective and automatic factors related to increased *motivation* for handwashing comprised an increased understanding of the link between handwashing and good health that encourages positive feeling about handwashing, disgust caused by realising that STH infection is linked to ingesting faecal matter; the use of relatable child characters that triggered a feeling of nurture. The main barrier identified was the inconsistent availability of water and soap.

## Capability

Two significant themes emerged from the analysis related to handwashing capability: perceived improvements in handwashing skills and techniques and reported improvements in knowledge about when and how to wash hands.

*Handwashing skills:* During handwashing demonstrations, all volunteers were able to follow correct handwashing steps. Participants reported to have mastered handwashing because they had participated in practical handwashing sessions.

> The teacher took us to the handwashing stations and showed us how to wash hands. We were told that when you come from the toilet you wet your hands, apply soap, then scrub (FGD older boys, school 2).

Both teachers and students believed that the acquired skills were important in facilitating handwashing.

*Handwashing lessons and educational messaging:* Both children and teachers indicated that teaching about handwashing and the use of educational messaging had greatly improved children's knowledge about how and when to wash hands. All children in both the FGDs and FPIs recalled receiving at least two lessons specific to handwashing in addition to the general sanitation and hygiene education provided within their normal school curriculum. During the interviews and FGDs we asked students the following question: *What happens if you don't wash your hands?* Students could explain the link between hand hygiene and infections. The quote below is an example of common stories throughout the students' interviews.

> If I do not wash my hands, I get diseases especially worms. For example if I eat even a fruit without washing my hands that means I am eating dirt and I will get worms and stomach ache. So whenever you eat without washing hands you get worms (FPI, young girl, school 3)

Students also discussed their own behaviour related to the key times to wash hands with soap (after using the toilet and before eating), as illustrated in the following quote.

> Before the project came … We did not always wash hands after using the toilet. But now were are motivated because we have been educated on the importance of handwashing so we now wash our hands always (FPI, young girls, school 1)

Knowledge about handwashing and key times to wash hands with soap were consistent across all interviews and FGDs. We noted no meaningful differences between students in rural and urban areas or differences between older and younger children regarding handwashing knowledge.

## Opportunity

The Mikono Safi intervention influenced opportunity for handwashing in two important ways—changing the physical opportunity for handwashing and altering the social environment related to handwashing at schools.

*Physical Opportunity:* Teachers and students alike indicated that availability and location of the handwashing facilities were important facilitators of handwashing. Teachers in particular noted that every child had to pass handwashing stations on their way to and from the latrine. Handwashing stations were visible and difficult to ignore.

> Given the location of the handwashing stations… on the way out of the toilet, everybody who uses the toilet can see them on their way out…but you know children like to play with water, so making water and soap available for children makes handwashing part of a game (IDI, Teacher 2, school 3)

> …We wash our hands at the handwashing stations near the toilet. Before they installed the handwashing tanks we never used to wash hands because we did not have water to wash hands but these days we wash our hands because they have put the tanks. But in the past we never used to wash hands (FPI, younger girls, school 3)

Children in the FGDs indicated that the improved handwashing facilities were not only easier and more convenient to use when compared with the tippy-taps that the school was using before but also the tanks contained much more water allowing a larger number of students to have access to water for washing hands.

> …Before the installation of the handwashing tanks we were using tippy-taps to wash hands and we had only few of them, the water would get finished quickly. But they were even difficult to use. With the current handwashing tanks you go there and open the tap and wash hands. The tanks are big and many students are able to wash hands using water in the tanks. (FGD, older boys, school 1)

Beyond the physical availability of handwashing facilities, the most important facilitator to handwashing was the consistent provision of supplies (water and soap). In some schools, especially schools with a large student population, it was not uncommon for water to get used up early in the day—particularly during break time when the majority of the children used the toilet. Schools were able to mitigate these interruptions in water availability in various ways including creating rosters for fetching water. In schools that had well supervised and well implemented class rosters for fetching water more children indicated that water was consistently available at the handwashing station.

> …there are times when water gets finished, but in our school we have a teacher and students responsible for health and hygiene. So when water is finished from the tanks we inform the teacher or student leader and they send a group of students to bring water (FPI, older girls, school 2)

Water availability, however, was often determined by factors outside of the school's direct control. All schools experienced disruptions in water supply to varying degrees. In three of the four schools in the study, this was a problem that occurred during the morning break when the majority of the students used the toilets and during the dry season in which the majority of schools in the region experience water shortages. In one school this problem occurred multiple times during a week. During periods of limited availability, teachers needed to interrupt classes for certain groups of students in order for them to fill the tanks. Often, teachers were unwilling to interrupt classes, especially in schools that did not have a well-established class roster for refilling water tanks. Lack of water had a negative influence on handwashing behaviours in general, particularly if water was unavailable for more than 1 day in a week.

In general, both students and teachers reported that soap was not consistently provided for washing hands. This was primarily an issue of distribution rather than of availability. At one school, for example, there was no soap available for students at the handwashing station when FGD participants were asked to demonstrate behaviours. Students noted that they had reported this to the teacher responsible earlier but in vain. However, when the moderator sent a student to the head teacher's office, the soap was provided. Similarly, discussions with school administrators indicated that, when provision of soap was inconsistent, it was usually available at the store.

> There are a number of times during the week when soap is not available at the handwashing station. This can happen like three times in a week. If the soap is not at the handwashing station we usually go to the Head teacher's office to get the soap (FGD, younger girls, school 3)

> There are times when the soap is not available because the teacher responsible did not check or the students have not reported. This happens sometimes but that usually is for a few hours in a day, not the whole day (IDI, Teacher 1, school 2)

*Social opportunity:* Data suggest that the Mikono Safi intervention had created school communities that valued handwashing and that teachers were willing to support handwashing activities through teaching and reminding children to wash hands, supervising children to fill the water tanks, and ensuring that soap was available. Students were also willing to participate in activities that ensured water and soap were available at the handwashing stations:

> If I came out of the toilet and there is no water I will pick a bucket get water from the main reserve tank and fill the handwashing tanks so that other children and I can wash our hands (FGD, Older girls, school 2)

> Today I went to the toilet in the morning but there was no soap at the handwashing station at that time. I reported to the student in charge of sanitation and s/he promised to inform the teacher that there was not soap at the handwashing station (FPI, older girls, school 1)

Students, especially the youngest ones, were keen to report colleagues that did not wash hands after using the toilet. Teachers described experiences where students would report to them other students who did not wash hands.

> Certainly, there are changes in children's handwashing behaviour. I am a teacher for the pre-primary class (the youngest group in school). What I have observed is that these days these young ones report colleagues that do not wash hands after using the toilet. If one of them used a toilet and did not wash hands the others will come running to you to report him or her; they will come and tell you 'teacher, so and so did not wash his/her hands after using the toilet'; they give you such reports (IDI, teacher, school 2)

## Motivation

*Automatic motivation/Emotional triggers*: Data suggest that the intervention was successful in reaching targeted emotional drivers of disgust, fear and nurture in children. In particular, many children described Muta—the boy character depicted in the educational materials as a boy who did not know how to wash his hands—as repulsive and disgusting. On the other hand children found Koku (portrayed as a hygiene loving child) attractive and every child interviewed indicated that he or she would like to be like Koku. Koku was described as clean and smart, but also kind and willing to teach a fellow student who may not have knowledge of handwashing yet.

> I feel bad about Muta's behaviour; the fact that he did not wash his hand makes me feel disgusted…. but we like Koku because she was clean and she knew how to wash hands in order to protect herself from disease. Everyone would like to be like Koku, she is clean, smart and she likes helping others (FGD, older girls, school 4)

> … the teachers told us that if you use the toilet but you do not wash your hands and you go and eat anything you may be eating feces. So when you think about that you remember to wash your hands and you will remind all your friends to wash hands (FGD, older girls, school 1)

*Reflective motivation:* Although introducing class rosters for fetching water was not part of the original intervention package, two of the four schools used rosters to ensure water was available at the handwashing facilities. For some schools this activity was led by students lead and for others by a teacher. For example, in schools that had active students' health clubs, members of these clubs were charged with the responsibility for supervising all health related activities in school. Hence, schools which had active student health clubs developed the roster for health activities for each day of the week.

Students in schools that had student leaders involved in taking decisions about replenishing water and soap were more motivated to take action to ensure they have materials to wash hands. When children believed that they had power to influence their own handwashing practices—they were willing to take an active role in activities that would facilitate handwashing among themselves and others.

> The students have received the handwashing lessons very well. They wash hand but they also encourage others to wash hands. For example if they saw a colleague who is dirty they encourage him/her to wash hand in order to avoid being labeled Muta, the character depicted in some of the education materials of a dirty boy… (IDI, teacher 1, school 3)

### Changes over time

Results from the two rounds of data collection were compared to detect possible differences in terms of perceived barriers and facilitators related to capability, opportunity and motivation for handwashing. We did not find any differences. In both round one and two, students demonstrated good knowledge and skills for handwashing. Similarly, the opportunity for handwashing remained largely unchanged between the data collection points at round one and two. The installed handwashing facilities remained well maintained in all four schools. The schools continued to experience disruptions in water supply, and as before success in mitigating such disruptions varied from school to school.

**Table 3** Capability, opportunity and motivational determinants of HWWS as reported by the participants

| Determinant | | Facilitators | Barriers |
|---|---|---|---|
| Capability | Psychological | Knowledge about the relationship between HWWS and STH<br>Knowledge of key moments for hand hygiene | |
| | Physical | Skills developed through HWWS demonstration and rehearsal | |
| Opportunity | Social | Teachers support and encourage HWWS<br>Peers who support HWWS | |
| | Physical | Improved quality, quantity and location of HW stations | Inconsistent availability of water<br>Inconsistent availability of soap at HW stations |
| Motivation | Automatic | Emotional response (disgust/fear) triggered through messages and content<br>Relatable characters inspiring feelings of nurture among older students<br>Reminders (nudges) that reinforced handwashing habits | |
| | Reflective | Proactive student engagement in water collection and soap provision | |

HW, handwashing; HWWS, handwashing with soap; STH, Soil-transmitted helminth.

Table 3 summarises capability, opportunity and motivational determinants of HWWS based associated with the Mikono Safi Intervention based on student and teacher reported data.

## DISCUSSION

Our study set out to describe the barriers and facilitators to handwashing behaviour from the perspectives of teachers and students participating in the Mikono Safi intervention trial. Our results show that participants associated the intervention with improved skills and knowledge. Children's description of the intervention matched with the targeted emotional drivers of behaviours. Both students and teachers described changes in the social environment of the school that fostered better HWWS behaviours. However, the physical opportunity for handwashing was mixed. When both water and soap were available, the new handwashing stations increased the ease and convenience of handwashing among students. On the other hand, we noted that schools reported on-going challenges with water availability and that the provision of soap at handwashing stations—even when available at the school—remained inconsistent.

Our data suggest that children receiving the Mikono Safi intervention had high capability for HWWS. Students could both describe and demonstrate proper hand hygiene and reported knowing when and why hands should be washed with soap. Although insufficient on its own, knowledge may be a necessary precursor for HWWS.[27] It is important to note that education and health messaging included in the Mikono Safi intervention were not abstract—they were focussed around the risk of preventing intestinal worms. Studies in India have shown that providing individuals with specific and actionable health information can improve WASH behaviours.[27] Having lessons that focus on hand hygiene and its risk with specific health outcomes in addition to regular sanitation and hygiene education may be a promising option for facilitating handwashing behaviour among school children.

The qualitative data from students and teachers suggest that the Mikono Safi intervention was associated with the targeted emotional drivers of nurture, fear and disgust. In the student data, there were rich stories of children describing Muta's poor hygiene as repulsive and disgusting. Students noted that they often washed hands to avoid being labelled 'Muta' by their colleagues. Studies have shown how motivational drivers can be successful in changing hygiene behaviours,[28] although only a limited number of studies have explored handwashing motives related to children. The SuperAmma study in India[28] demonstrated significant improvement in hygiene behaviours based on messaging that targeted motivational drivers in isolation from health education. Our study shows how these emotional drivers can be combined with targeted health education to influence both motivation and capability.

Our findings suggest that the nurture motive was particularly salient to students. Students spoke fondly of the fact that Koku (the female student appearing in education materials) helped her friends out; and both teachers and students discussed the ways that the school had provided a supportive environment for handwashing. The salience of the nurture motive in our study population is consistent with other studies of students handwashing. For example, a study in Bangladesh that used covert video cameras to record handwashing behaviour found numerous instances of older children helping younger children washing hands and modelling positive behaviours for one another.[29] Students in our study responded positively to the nurture messages, suggesting that positive peer pressures are a potential avenue for improving HWWS in schools.

Making handwashing facilities visible, difficult to ignore but also convenient, were among the attributes mentioned as important by both teachers and students. However, beyond the physical positioning and convenience, the most important facilitator to handwashing was consistent provision of water and soap. Although all schools experienced disruptions in water supply especially at peak hours, some schools that were able to implement a well supervised class roster for fetching water succeeded in avoiding interruptions in water availability at handwashing stations. Similarly students and teachers suggested that inconsistency in the availability of soap at the handwashing stations was largely an issue of internal distribution rather than of external logistics. Our data suggests that while the Mikono Safi intervention was successful in changing capability and motivation, the opportunity for HWWS may not have been sufficiently improved in some schools, either due to external factors such as the dry season, or to internal factors such as the lack of class duty rosters for the collection of water or soap.

Our study has some limitations : first, this was a qualitative study involving a small sample of schools, teachers and pupils, and therefore our results may not be generalisable to other intervention schools. However, our schools were purposively selected to reflect the general diversity of schools included in the Mikono Safi study and are similar to most schools in the Kagera region. Second, findings are primarily based on individual interviews and FGDs. Data from direct observation of students' behaviour was not collected during the qualitative study. However, FGDs, FPIs with students and IDIs with teachers proved to be well suited methods to examine barriers and facilitators to appropriate handwashing behaviour, and produced consistent results. We also do not have information on drivers of behaviour before the intervention was implemented and were not able to interview students in enrolled control groups and therefore can only draw associations between the intervention and students reported determinants of handwashing. Students and teachers were aware of our affiliation with the larger Mikono Safi intervention trial and courtesy bias may have

influenced both teachers and students to provide answers they assumed we wanted to hear. Reports on changes in behaviour should therefore be interpreted with caution. However, our study explored the drivers of behaviour rather than the behaviours themselves. We also found high consistency in responses from both students and teachers, and respondents were forthcoming about challenges related to sustained soap provision at schools.

The COM-B Framework provided a prudent way to explore the success of the Mikono Safi intervention in addressing the barriers and facilitators for handwashing in our study population. Our analysis was based on drivers and facilitators among students. Viewed from an institutional perspective, several of our key findings could relate to other COM-B determinants. For example, consistent provision of soap from the institution's perspective may be a factor of reflective motivation—schools may not have prioritised soap distribution or they could have made conscious decisions to withhold materials from students out of budget or supply concerns. Given the important role in intervention compliance in several school-based WASH interventions,[11 14] more theoretically informed research is needed to unpack the drivers of institutional factors that contribute to behavioural outcomes.

In conclusion, the Mikono Safi intervention was associated with high motivation and capability for handwashing, but ensuring the opportunity for handwashing remained a challenge at some schools, especially with respect to a consistent availability of water. For such schools a much stronger multi-sectoral collaboration between the departments for education and water may be required. Results from the forthcoming trial will allow us to understand the impact this intervention had on child health outcomes. Future research focussed on hygiene and hygiene behaviour change should further explore the ways in which motivational and educational messaging can change specific determinants of behaviour; new intervention modalities that promote school-level adherence to routine provision of basic supplies need development and testing in resource-scarce environments like Tanzania

**Acknowledgements** We thank John Robert and Winnie Muriba for assisting with data collection and transcription of the interviews and focus group discussions. We are grateful to the parents for allowing their children to take part in the study and the school administration of the 4 schools for giving us permission to talk to students and teachers. We are grateful to the participating students and teachers for their time and contributions to the study.

**Contributors** EO conceived the study and was the investigator with overall responsibility for the management of the study. EO and RD were responsible for the original design and protocol. SK, HG, SK, contributed to the study design methodology and obtaining funding. KM and OM supervised fieldwork and reviewed the draft manuscript. EO analysed the data and wrote the first draft of the manuscript. RD, HG, SK and SK provided a critical review for important intellectual content. All authors were involved in the final version of the manuscript and approved the manuscript for submission.

**Funding** This study was funded by the Department of International Development (DFID) of the United Kingdom (UK) through the SHARE Consortium programme, based at the London School of Hygiene and Tropical Medicine (LSHTM).

**Disclaimer** The views expressed are those of the authors and not necessarily those of the funder.

**Competing interests** None declared.

**Patient consent for publication** Not required.

**Ethics approval** Protocol, tools and consent forms were reviewed by the National Health Research Ethics Review Committee of Tanzania (approval number: NIMR/HQ/R.8a/Vol.IX/2665) and the Ethics Committee of the London School of Hygiene and Tropical Medicine (LSHTM Ethics Ref: 14539).

**Provenance and peer review** Not commissioned; externally peer reviewed.

**Data availability statement** Data are available upon reasonable request.

**ORCID iD**
Elialilia Okello http://orcid.org/0000-0002-0507-1815

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
