## [Reviewer comments · BMJ Open]

ARTICLE DETAILS

TITLE (PROVISIONAL)	Factors perceived to facilitate or hinder handwashing among primary students: a qualitative assessment of the Mikono Safi intervention schools in NW Tanzania
AUTHORS	Okello, Elialilia; Kapiga, Saidi; Grosskurth, Heiner; Makata, Kenneth; Mcharo, Onike; Kinungh'i, Safari; Dreibelbis, Robert

VERSION 1 – REVIEW

REVIEWER	Le Thi Thanh Xuan Institute for Preventive Medicine and Public Health, Hanoi Medical University, Vietnam
REVIEW RETURNED	03-May-2019

GENERAL COMMENTS	To provide research question and ethical approval, informed consent
---

REVIEWER	Ruby Biezen University of Melbourne, Australia
REVIEW RETURNED	15-May-2019

GENERAL COMMENTS	Manuscript Number: bmjopen-2019-030947 Title: Factors perceived to facilitate or hinder handwashing among primary students: a qualitative assessment of the Mikono Safi intervention schools in NW Tanzania Thank you for the opportunity to review this manuscript. This is an extremely important study and I commend the research team for this type of research and by doing so, hopefully will enable better hygiene and health outcome in Tanzania. While the study is of good quality, and the research sound, however, the manuscript may benefit from proof reading by a native English speaker as there were grammatical errors throughout. In addition, the results could be analysed more in depth, and utilise the COM-B model in more detail, instead of just 'reporting' the results. I would also suggest comparing the two time-frames to enhance your analysis, rather than just saying 'there was no difference between data collection (do you mean data analysis?) on page 22. I have provided some comments for the authors to consider. Furthermore, future research/directions and implication of your result need to be stronger, and perhaps include a sentence in the abstract to show your audience that you have considered the implication of this important piece of research. Some comments that the authors might like to consider:
--

	Abstract:  • P2, para 1: You mentioned implementation fidelity, and also in the end of introduction as part of your goal. That is the end of any further mention of implementation fidelity. Can you explain what this is, how this relates to your study and include in your discussion? • P2, para 5: The Mikono Safi interventions (or intervention?) was mentioned but not in the design. Suggest include a sentence of what this is in the design. • P2, para 5: 'HWWS' acronyms need to be spell out in the first instance. • P2, para 5: How does the participatory learning approach to hand hygiene education tied into the intervention? • As you used the COM-B model to analyse the data, suggest including this in the abstract. Strengths and limitations of this study:  • Need to expand 'FGDs' and 'FPI' in the first instance. • Consider changing 'with' with 'by' in the second point. • How were the contact sustained with students and teachers during the two rounds of data? Suggest making this clear in the methods section. • What do you mean by social desirability bias? Introduction:  • P5, para 1: I thought respiratory tract infections are more prevalent worldwide then soil-transmitted helminth? • Can you explain what MDA programs are and how this relate to your study? Is MDA specific to STH infections? • P5, end of para 1: "This is the rationale underlying ... The trial will assess..." should be at the end of your introduction, and the reasons leading to the aim of this study. • Your introduction needs to be rearranged: You started off with a specific parasitic infection, then move onto the rationale of your study, then you go back and talk about general infections and health outcomes. Is the goal of your study to reduce STH infections or is that part of the Mikono Safi study? Perhaps explaining the Mikono Safi study would explain how this ties together. As is, your introduction seems scattered. • P6, para 3: This should be part of your methods section, not introduction. How you used COM-B to analyse your data throughout the manuscript was not clear. Methods  • Study setting and intervention development: A lot of this section, especially the Mikono Safi intervention, could be included in the introduction as part of your larger study. This would fit nicely in the introduction: after explaining the problem (transmission of infections), what work had been done in recent times and the gaps identified, then the rationale underlying the Mikono Safi intervention, and your study. Then you just need to include study design, sample, data collection, data analysis etc in your methodology. • Please include whether Ethics were acquired, and how consent was obtained? • Data collection for round 1 and round 2 was not clear, suggest a diagram with timeline might help in explaining the sequence of events. • Can you explain the purpose of round 1 and round 2 data collection? How did you analyse these data? Did you compare and contrast between the two rounds? • What was the rationale of using FPIs instead of FGD in round 2?
--	--

	 • P12 para 1: 'other stakeholders', who were they? How did you involve them in the study? What did you ask them to do? • It is unclear how you used the COM-B model to serve as a guide to your data analysis. It needed more explanation to demonstrate in your results that the data was analysed using this model, ie. What were the advantages of using COM-B? How did the results benefit from using this model? What did the analysis tell you in terms of change in the participants' behaviour? • Were the interviews and FGDs conducted in Kiswahili and transcribed in Kiswahili? • There seemed to be only questions for friendship pair interview guide for students – were the questions the same for the focus groups in round 1? This is really important, as it determined how you compared the data between round 1 and round 2 during data analysis. Results  • Were there any demographic differences between the two rounds? i.e. Age of the teachers, students etc? Did you interview the same teachers in round 1 and 2? Were there students that were involved in both round 1 and round 2? Suggest including a table to better display the demographic data. • P14, para 2: suggest taking out 'the' in 'We defined capability as the having...' • P14, para 2: Should 'improvements in hand washing skills' defined as 'capability'? or improvements as a result of knowledge? • P14, para 3: first sentence – this is an incomplete sentence. • P14, para 3, line 3: suggest rewriting this sentence to 'Participation in hand washing demonstrated that students were able to acquire the necessary hand washing skills'. • P15, second quote, missing school number. • In general, the results could be better analysed and provided more depth. This can be achieved by comparing and contrasting round 1 and round 2 data, and aligning with the COM-B model. Quotes could be reduced and shortened, and only select ones that were crucial to demonstrate your point; it is currently very long. • Page 19, last para: repetitive, same as previous page • Perhaps a diagram is needed to demonstrate how these themes fit into the COM-B model, and how they impact and change behaviour. • P22: Changes over time – how do you know there was no difference in terms of capability, opportunity and motivation between round 1 and 2 if there were no in-depth analysis between the two rounds? In order to demonstrate this, you will need to provide evidence (as in analysis to compare these two rounds), and include in the methodology how you did this. Discussion:  • P23, para 2: extra open bracket in citation 36 • P23, para 2, line 7: which studies? Suggest discussing some of these 'previous' studies and how they relate to your study and what they showed. • P23, para 3: Nurture was mentioned once in the results section under motivation, in conjunction with disgust and fear. Suggest you either discuss nurture with disgust and fear, or highlight nurture in the results so you can single this out in the discussion. • P24, para 2: I'm unsure whether these are 'attributes'. Suggest changing this to "...were among important elements mentioned by both teachers and students."
--	--

	 • Limitation: study conducted in Tanzania, may not be generalisable in other countries • Overall, discussion is heading in the right direction, however, more depth is needed, and suggest to be as succinct as possible. Conclusion:  • Was the COM-B framework only used in the qualitative component of the study or was it used for the whole research program? • What are the implications of the study? What are your recommendations? Appendix:  • Missing focus group questions. This is a very important and valuable study; I hope my comments helped.
--	---

REVIEWER	Dr. Emmanuel Appiah-Brempong Kwame Nkrumah University of Science and Technology, Ghana
REVIEW RETURNED	09-Jul-2019

GENERAL COMMENTS	COMMENTS TO AUTHORS Let me congratulate authors for taking time to research into such an important phenomenon, especially in a developing country setting where data generation is usually not a straight-forward process. I find the following comments useful for enriching the content of the manuscript. Title: "Factors perceived to facilitate or hinder handwashing among primary students..." "Proper handwashing" appears more appropriate because that is what is being promoted by the public health movement across the globe. Abstract Should be revised on the basis of the corrections made within the manuscript Introduction "...the mechanisms through which interventions change behaviours remains underexplored. Further, few studies examine intervention feasibility and acceptability from a child's perspective." Authors appear to be emphatic with the statement above. Can they explain the basis for such an assertion? For example, was an extensive literature search conducted to identify published/unpublished works on mechanisms through which interventions change behaviours? ...and also intervention feasibility and acceptability from a child's perspective? If so, which databases were consulted? (in the case of published electronic materials). Which search terms were used?, etc. Authors do not state clearly the precise objective of the paper and a justification of it within the introduction section. For example, later on in the discussion section, I find authors state that the "study was set out to describe the barriers and facilitators to handwashing behaviour from the perspectives of teachers and students" but this is mentioned nowhere in the introduction section. Methods "The sessions were delivered to complement the existing science curriculum which covered general body hygiene and sanitation with very limited emphasis to handwashing".
---

	“Some of the intervention schools, especially the rural based schools did not have any form handing washing facilities while urban based schools had basic tippy-taps”. Check the correct the grammar and sentence flow of the above sentences. “At each school, discussions were held with teachers to ensure soap was reliably made available for students to wash hands, and teachers were trained on how to implement handwashing lessons”. An elaborate description of this training will be useful for the purpose of replicating this study elsewhere. For example, who conducted the training? What is the qualification of the trainer? What were the objectives of the training sessions? What was the duration of the training? Were all teachers trained? If not, what criteria informed the selection of teachers? And what strategies were used to ensure that training was equally done for all intervention schools? “During the handwashing education sessions, teachers guided students to practice a correct way of washing ones”. ...correct way of washing hands?... Please read the entire manuscript carefully and correct all such errors. Study Design, Sample and Data Collection Methods Any justification for the choice of sampling technique? I do not think your justification is merely to have urban and rural schools. Also, what was the rationale for deciding to blend urban schools with rural schools? Any justification for the 2 months post intervention assessment? and the 8 months post intervention assessment? “To assess students’ handwashing skills, at the end of each FGD, the moderator walked with the participating students to the handwashing stations for students to demonstrate how they wash hands”. What tool was used by moderators for assessing proper handwashing skill of students? What was the numerical size of each FGD session? You also need to report on ethical considerations Results “Handwashing skills: Students indicated in both FGDs and FPIs that because they had been involved and had taken an active role in the learning about HWWS. Participation in the handwashing demonstrations they were able to acquire the necessary handwashing skills”. Check and reword sentences “Students understood the link between hand hygiene and infections. They also knew the key times they were required to wash hands” All students? Or some students? How did you test for understanding? Discussion The bit on “barriers and facilitators to handwashing behaviour from the ...” does not appear to reflect significantly in the results
--	--

	section. It may be good revisiting the results section and enriching it further to reflect the above. “Our health messaging also had an impact on the emotional drivers of hygiene behaviours.” How do you substantiate the above? Since you assessed impact from the perspective of the children and teachers, you need to be cautious about emphatic statements like this. “Data suggest that we were successful in triggering emotional responses in children through our information campaigns”. Which precise data and in what way does it suggest this? and from whose perspective? “Our data suggests that while the Mikono Safi intervention was successful in changing capability and motivation”. Which precise data and in what way does it suggest this? In addition to the limitations stated, it is worth stating that since the study relied largely on self-reported data, there is the risk of reporting bias due to the obvious tendency to provide socially desirable responses. Conclusion “In general, we found that the intervention was successful in improving motivation for handwashing and capability for handwashing.....” This conclusion raises questions considering the fact that the study adopted a qualitative approach. You appear emphatic even though you did not seek to determine a causal relationship (which apparently does not fall within the remit of a qualitative study). Furthermore, you have earlier indicated that success was assessed from the perspective of children, who are obviously prone to providing socially desirable responses. Also, I do not find the use of the phrase “In general” to be a scientific language. What precisely do you mean by that? I think your conclusion needs to be re-written.
--	--

VERSION 1 – AUTHOR RESPONSE

Reviewer: 1

1. To provide research question and ethical approval, informed consent

The study aim has been added on page 6 as follows: “The goal of our study was to explore the extent to which the Mikono Safi intervention influenced the determinants of children’s handwashing behaviour”. A section describing the ethical approval process has been included on pages 11 and 12

Reviewer: 2

2. While the study is of good quality, and the research sound, however, the manuscript may benefit from proof reading by a native English speaker as there were grammatical errors throughout.

A native English speaker within our team has helped to revise the wording of the text with regards to grammar and style. This has led to substantial changes of the text without changing the factual content.

3. In addition, the results could be analysed more in depth, and utilise the COM-B model in more detail, instead of just 'reporting' the results.

Additional text has been added, based on an amended analysis utilizing the COM B model. For example, we have reviewed the results section on pages 12 to 20. We have also included a table (table 3 on page 22) showing how our results fit in with the COM-B model.

4. I would also suggest comparing the two time-frames to enhance your analysis, rather than just saying 'there was no difference between data collection (do you mean data analysis?) on page 22.

This has been done. Please see the revised text on page 20 which now reads "Results from the two rounds of data collection were compared to examine differences in terms of perceived barriers and facilitators related to capability opportunity and motivation for handwashing and there were no differences. In both round one and two, students demonstrated good knowledge and skills for handwashing. Similarly, the opportunity for handwashing remained largely unchanged between the data collection round one and two..."

5. I have provided some comments for the authors to consider. Furthermore, future research/directions and implication of your result need to be stronger, and

Thank you very much for these helpful comments which have been adopted wherever possible. A section on the implications of the results has been included on pages 25-26. This now reads as follows: "...the Mikono Safi intervention was associated with improved motivation and capability for handwashing, but ensuring the opportunity for handwashing remained a challenge at some schools, especially with respect to a consistent availability of water. For such schools a much stronger multi-sectoral collaboration between the departments for education and water may be required. Results from the forthcoming trial will allow us to understand the impact this intervention had on child health outcomes. Future research focused on hygiene and hygiene behaviour change should further explore the ways in which motivational and educational messaging can change specific determinants of behaviour; new intervention modalities that promote school-level adherence to routine provision of basic supplies need development and testing in resource-scarce environments like Tanzania"

6. Perhaps include a sentence in the abstract to show your audience that you have considered the implication of this important piece of research.

This has been added to the abstract.

Some comments that the authors might like to consider:

Abstract:

7. P2, para 1: You mentioned implementation fidelity, and also in the end of introduction as part of your goal. That is the end of any further mention of implementation fidelity. Can you explain what this is, how this relates to your study and include in your discussion?

We thank the reviewer for correctly noting that an assessment of implementation fidelity was not a research goal for our study. We have revised the manuscript stating that whilst previous studies often interpreted health outcomes through the lens of intervention fidelity and compliance (page 5), our

study addresses the mechanisms and the extent to which the intervention influence the determinants of children's behaviour (pages 5 and 6).

8. P2, para 5: The Mikono Safi interventions (or intervention?) was mentioned but not in the design. Suggest include a sentence of what this is in the design.

This has been revised to include a description of the intervention design on pages 6 and 7

P2, para 5: 'HWWS' acronyms need to be spell out in the first instance.

The acronym has been removed from the abstract.

9. P2, para 5: How does the participatory learning approach to hand hygiene education tied into the intervention?

Following the revisions of the abstract, the sentence on participatory learning has been deleted.

10. As you used the COM-B model to analyse the data, suggest including this in the abstract.

This has been included in the abstract on page 1

Strengths and limitations of this study:

11. Need to expand 'FGDs' and 'FPI' in the first instance.

The FGDs and FPI are now stated in full when used for first time on page 3. The acronyms are now introduced when first mentioned in the Methods section (page 9)

12. Consider changing 'with' with 'by' in the second point.

This has been revised. The whole text has been removed from page 3.

13. How were the contact sustained with students and teachers during the two rounds of data? Suggest making this clear in the methods section.

The sentence has been deleted during revisions. (Please note that whilst the intervention team stayed in contact with all schools throughout, there was no scientific need for the qualitative research team to keep in contact with individual participants between rounds of data collection).

14. What do you mean by social desirability bias?

We have removed this term from the text. Instead we now mention the possibility of courtesy bias in the Discussion section (page 25) that may have influenced both teachers and students to provide answers they assumed (the interviewer) wanted to hear.

Introduction:

15. P5, para 1: I thought respiratory tract infections are more prevalent worldwide then soil-transmitted helminth?

The sentence has been rephrased to state that soil transmitted helminth infections are among the most prevalent infections... A reference to support this statement has been added on page 4

16. Can you explain what MDA programs are and how this relates to your study? Is MDA specific to STH infections?

This has been revised to include information on page 4 on how the mass drug administration relates to our study.

17. P5, end of para 1: "This is the rationale underlying ... The trial will assess..." should be at the end of your introduction, and the reasons leading to the aim of this study.

This text has been removed. Instead we now describe in the Introduction section (page 6) that "The goal of our study was to explore the mechanisms by which and the extent to which the Mikono Safi intervention influenced the determinants of children's handwashing behaviour" The rationale leading to this objective is provided in the paragraph just before this sentence.

18. Your introduction needs to be rearranged: You started off with a specific parasitic infection, then move onto the rationale of your study, then you go back and talk about general infections and health outcomes. Is the goal of your study to reduce STH infections or is that part of the Mikono Safi study? Perhaps explaining the Mikono Safi study would explain how this ties together. As is, your introduction seems scattered.

We have re-arranged the introduction to improve the flow of the text, to make the objectives of the Mikono Safi intervention and trial clear, and to differentiate the objectives of the qualitative study more clearly from those of the larger intervention trial.

The introduction now introduces the burden of STH and school-based approaches to STH control, introduces new evidence suggesting that school-based WASH programs can help reduce reinfection rates, and then introduces the Mikono Safi intervention and trial (page 5).

19. P6, para 3: This should be part of your methods section, not introduction. How you used COM-B to analyse your data throughout the manuscript was not clear.

The introduction has been revised to improve the flow of the text (pages 4-7).

The description of the intervention has been removed from the Methods section and has been included as a separate subsection on page 6 because we feel that it will provide context and improve understanding for current qualitative study.

Methods

20. Study setting and intervention development: A lot of this section, especially the Mikono Safi intervention, could be included in the introduction as part of your larger study. This would fit nicely in the introduction: after explaining the problem (transmission of infections), what work had been done in recent times and the gaps identified, then the rationale underlying the Mikono Safi intervention, and your study. Then you just need to include study design, sample, data collection, data analysis etc in your methodology.

We thank the interviewer for this advice. A brief summary of the Intervention is now provided in the Introduction section on pages 6-7

21. Please include whether Ethics were acquired, and how consent was obtained?

A statement about ethical considerations and the consent process has been included on page 11-12.

22. Data collection for round 1 and round 2 was not clear; suggest a diagram with timeline might help in explaining the sequence of events.

A table describing the study time line, data rounds and methods of data collection has been included on page 9.

23. Can you explain the purpose of round 1 and round 2 data collection? How did you analyse these data? Did you compare and contrast between the two rounds?

This information has been provided on pages 8. The purpose of two rounds of data collection is to document whether there were changes over time with regards to observations on the determinants of hand washing behaviour. We compared the results from both rounds and did not find such changes (page 21, at the end of the Results section).

What was the rationale of using FPIs instead of FGD in round 2?

The rationale for this has been included on page 10. We used FGDs to explore students shared perceptions and FPIs to explore individual experiences

24. P12 para 1: 'other stakeholders', who were they? How did you involve them in the study? What did you asked them to do?

These stakeholders included 'regional and district education officers and school WASH coordinators' who had contributed to the design of the Mikono Safi intervention during a previously conducted formative research phase. A detailed description has been included in the main document on page 11.

25. It is unclear how you used the COM-B model served as a guide to your data analysis. It needed more explanation to demonstrate in your results that the data was analysed using this model, i.e. What were the advantages of using COM-B? How did the results benefit from using this model? What did the analysis tell you in terms of change in the participants' behaviour?

We used COM-B throughout the entire study, from conception to analysis. We have added a section on the Theoretical Framework in the beginning of the Methods section (pages 6 and 7). In general, we note that our research explored the key determinants of handwashing behaviour. COM-B provided us with a flexible tool for both the design and analysis of our results.

26. Were the interviews and FGDs conducted in Kiswahili and transcribed in Kiswahili?

All the data from teachers and students was collected in Kiswahili, transcribed in Kiswahili and translated into English after transcription. A statement about this has been included on page 11.

27. There seemed to be only questions for friendship pair interview guide for students – were the questions the same for the focus groups in round 1? This is really important, as it determined how you compared the data between round 1 and round 2 during data analysis.

Questions on FGD and FPI guides were largely the same except that in FGDs we focused on shared perceptions while in FPI we explored individual experiences. We have included a brief clarification of this on page 10.

Results

28. Were there any demographic differences between the two rounds? i.e. Age of the teachers, students etc? Did you interview the same teachers in round 1 and 2? Were there students that were involved in both round 1 and round 2? Suggest including a table to better display the demographic data.

Sixteen teachers were recruited to the study and they were interviewed twice. A table summarising participants' gender and age has been included on page 12.

29. P14, para 2: suggest taking out 'the' in 'We defined capability as the having...'

This has been revised.

30. P14, para 2: Should 'improvements in hand washing skills' defined as 'capability'? or improvements as a result of knowledge?

Knowledge and skills are both part of the larger capability construct in the COM-B framework, therefore we have included them under the same heading, as detailed on pages 13 and 14

31. P14, para 3: first sentence – this is an incomplete sentence.

This has been revised.

32. P14, para 3, line 3: suggest rewriting this sentence to 'Participation in hand washing demonstrated that students were able to acquire the necessary hand washing skills'.

This paragraph has been revised.

33. P15, second quote, missing school number

The number has been added on the quotation and page 14 to identify the school

34. In general, the results could be better analysed and provided more depth. This can be achieved by comparing and contrasting round 1 and round 2 data, and aligning with the COM-B model. Quotes could be reduced and shortened, and only select ones that were crucial to demonstrate your point; it is currently very long.

Further synthesis has been done to align our data with the COM-B model and some quotes have been deleted.

35. Page 19, last para: repetitive, same as previous page

Extensive revisions have been made to eliminate repetitions

36. Perhaps a diagram is needed to demonstrate how these themes fit into the COM-B model, and how they impact and change behaviour.

A table has been included on page 22 showing how our results fit into the COM-B framework and how barriers and facilitators across the three determinants interact to shape students' handwashing behaviour

37. P22: Changes over time – how do you know there was no difference in terms of capability, opportunity and motivation between round 1 and 2 if there were no in-depth analysis between the two

rounds? In order to demonstrate this, you will need to provide evidence (as in analysis to compare these two rounds), and include in the methodology how you did this.

This comment has been addressed on page 20 where we describe the comparison of findings between rounds. We beg to disagree that an additional in-depth analysis between the two rounds would have been required to determine whether observations from round 1 and round 2 differ or are identical.

Discussion:

38. P23, para 2: extra open bracket in citation 36

This has been revised.

39. P23, para 2, line 7: which studies? Suggest discussing some of these 'previous' studies and how they relate to your study and what they showed.

This has been revised to state how the previous studies relate to the current study on pages 23 and 24

40. P23, para 3: Nurture was mentioned once in the results section under motivation, in conjunction with disgust and fear. Suggest you either discuss nurture with disgust and fear, or highlight nurture in the results so you can single this out in the discussion.

Details on 'nurture' have been provided in the Results section on page 19 and in the Discussion section on page 23.

41. P24, para 2: I'm unsure whether these are 'attributes'. Suggest changing this to "...were among important elements mentioned by both teachers and students."

This has been revised on page 24 to read as follows: "Making handwashing facilities visible, difficult to ignore but also convenient, were among the attributes mentioned as important by both teachers and students"

42. Limitation: study conducted in Tanzania, may not be generalisable in other countries

This has been added to the text on limitation on page 3 and in the Discussion section on page 25.

43. Overall, discussion is heading in the right direction; however, more depth is needed, and suggest to be as succinct as possible.

The discussion section has been expanded in line with this concern.

Conclusion:

Was the COM-B framework only used in the qualitative component of the study or was it used for the whole research program?

We used the COM-B framework to inform the design and analysis of this qualitative study. The Mikono Safi intervention was informed by the Behaviour Centred Design process as described on page 6

44. What are the implications of the study? What are your recommendations?

The implications of the study are now described in detail in the Discussion section (pages 22 – 25) and have been summarised on page 26.

Appendix:

45. Missing focus group questions.

We used the same questions for both FGDs and FPIs. Except that we used FGD to assess shared perceptions so a normative approach was used when asking this question while the FPI assess individual perceptions and experiences (the guide has been included as a supplementary file)

This is a very important and valuable study; I hope my comments helped.

We appreciate the reviewers' valuable comments. They have been very helpful in improving the quality of the manuscript.

Reviewer: 3

46. Let me congratulate authors for taking time to research into such an important phenomenon, especially in a developing country setting where data generation is usually not a straight-forward process. I find the following comments useful for enriching the content of the manuscript.

We appreciate the comments which have been extremely useful.

47. Title: "Factors perceived to facilitate or hinder handwashing among primary students..." "Proper handwashing" appears more appropriate because that is what is being promoted by the public health movement across the globe.

We appreciate the comment, although in the paper we have reported some observations on handwashing demonstrations done during FGDs, we did not collect systematic observational data on handwashing practices, we would like to keep the original title as it is.

Abstract

48. Should be revised on the basis of the corrections made within the manuscript

We have revised the abstract following the revisions done in the main text.

Introduction

"the mechanism through which interventions change behaviours remains underexplored. Further, few studies examine intervention feasibility and acceptability from a child's perspective." Authors appear to be emphatic with the statement above. Can they explain the basis for such an assertion? For example, was an extensive literature search conducted to identify published/unpublished works on mechanisms through which interventions change behaviours? ...and also intervention feasibility and acceptability from a child's perspective? If so, which databases were consulted? (in the case of published electronic materials). Which search terms were used?, etc.

This text has been revised to be less emphatic. In the Introduction section we just allude to the fact that "While school-based WASH interventions in low- and middle-income countries have been examined with regards to a wide range of health and educational outcomes, ...these studies have shown that the impact on such outcomes is inconsistent." Five references are provided to support this conclusion. We continue by arguing that "... for school-based hygiene interventions to result in health

or educational impacts, they must also successfully change students' hygiene behaviours." (Page 5). The factors determining such behaviour is the focus of this qualitative study.

49. Authors do not state clearly the precise objective of the paper and a justification of it within the introduction section. For example, later on in the discussion section, I find authors state that the "study was set out to describe the barriers and facilitators to handwashing behaviour from the perspectives of teachers and students" but this is mentioned nowhere in the introduction section.

This has been revised and the objective of the study has been clearly stated on page 6 as follows: "The goal of our study was to explore the mechanisms by which and the extent to which the Mikono Safi intervention influenced the determinants of children's handwashing behaviour".

Methods

50. "The sessions were delivered to complement the existing science curriculum which covered general body hygiene and sanitation with very limited emphasis to handwashing". "Some of the intervention schools, especially the rural based schools did not have any form handing washing facilities while urban based schools had basic tippy-taps". Check the correct the grammar and sentence flow of the above sentences.

The text has been revised by a native English speaker within the team and issues related to grammar and writing style have been addressed.

51. "At each school, discussions were held with teachers to ensure soap was reliably made available for students to wash hands, and teachers were trained on how to implement handwashing lessons". An elaborate description of this training will be useful for the purpose of replicating this study elsewhere. For example, who conducted the training? What is the qualification of the trainer? What were the objectives of the training sessions? What was the duration of the training? Were all teachers trained? If not, what criteria informed the selection of teachers? And what strategies were used to ensure that training was equally done for all intervention schools?

This has been briefly described on pages 6 and 7. Please note that a detailed description of the intervention is not the objective of this manuscript. The detail will be published in a separate paper that will document the design of the intervention (and the results of the quantitative baseline survey).

52. "During the handwashing education sessions, teachers guides students to practice a correct way of washing ones"...correct way of washing hands?...Please read the entire manuscript carefully and correct all such errors.

This section has been revised and spelling and grammatical errors corrected.

Study Design, Sample and Data Collection Methods

53. Any justification for the choice of sampling technique? I do not think your justification is merely to have urban and rural schools. Also, what was the rationale for deciding to blend urban schools with rural schools?

Our sampling strategy aimed to obtain a representative cross-section of schools and participants of the 16 primary schools and their students that participate in the main trial. We realise that this is difficult, given that for this qualitative study we could only include 4 schools of the 16 schools. Nevertheless, the sample mirrors to the extent possible the distribution of schools and participants with respect to geographical location, baseline epidemiology (high versus low pre-trial STH infection prevalence), age and gender distribution of students and roles of participating teachers.

54. Any justification for the 2 months post intervention assessment? and the 8 months post intervention assessment?

Justification for 2 months: The intervention needed to be fully operational for us to be able to assess the intervention's influence on determinants of behaviour

Justification for 8 months: a time of at least 6 further months was needed to make a meaningful comparison between data from the first and the second round of data collection.

55. "To assess students' handwashing skills, at the end of each FGD, the moderator walked with the participating students to the handwashing stations for students to demonstrate how they wash hands". What tool was used by moderators for assessing proper handwashing skill of students?

Field researchers were trained to assess handwashing skills by checking whether students used water and soap, applying an appropriate time needed to wash both hands and all fingers. To support them, they used a brochure which described the steps of correct handwashing.

56. What was the numerical size of each FGD session?

The size of FGDs ranged from 6 to 10 participants as implied by the text on page 12 on FGDs.

57. You also need to report on ethical considerations

A statement on ethical considerations has been included on page 13 (please see equivalent responses to comments from reviewers 2 and 3.

Results

58. "Handwashing skills: Students indicated in both FGDs and FPIs that because they had been involved and had taken an active role in the learning about HWWS. Participation in the handwashing demonstrations they were able to acquire the necessary handwashing skills". Check and reword sentences

This has been revised.

59. "Students understood the link between hand hygiene and infections. They also knew the key times they were required to wash hands". All students? Or some students? How did you test for understanding?

In this qualitative study we did not collect data on exact numbers of respondents to specific questions asked during focus group discussions or in interviews of students and teachers. As statement such as the one given above implies that most or all students had met the observation. Understanding was tested by asking questions to check whether students could recall information that had been taught during education sessions.

Discussion

60. The bit on "barriers and facilitators to handwashing behaviour from the ..." does not appear to reflect significantly in the results section. It may be good revisiting the results section and enriching it further to reflect the above.

We appreciate the comments. The text on barriers has been expanded accordingly on pages 15 and 16

61. “Our health messaging also had an impact on the emotional drivers of hygiene behaviours.” How do you substantiate the above? Since you assessed impact from the perspective of the children and teachers, you need to be cautious about emphatic statements like this.

We appreciate the comment. This has been revised on page 23 and 24 and now reads as follows: “Our education package also had an impact on the emotional drivers of hygiene behaviours. Posters and information went beyond just health messaging and included elements of disgust fear and nurture. Data suggest that we were successful in triggering emotional responses in children through our information campaigns..... Our intervention specifically targeted three emotional drivers: nurture, fear and disgust. However, nurture was seemingly the most important emotional driver that we were successful in targeting with intervention activities. Students spoke fondly of the fact that Koku (the female student appearing in education materials) helped her friends out; and both teachers and students discussed the ways that the school had provided a supportive environment for handwashing. ...Students in our study responded positively to the nurture messages, suggesting that positive peer pressures are a potential avenue for improving HWWS in schools.

62. In addition to the limitations stated, it is worth stating that since the study relied largely on self-reported data, there is the risk of reporting bias due to the obvious tendency to provide socially desirable responses.

We have included this under limitations in the Discussion section (page 25). In our manuscript, this kind of bias is called ‘courtesy bias’. However, please note that our self-reported data was not only about behaviours but also their drivers which may be less prone to social desirability bias.

Conclusion

63. “In general, we found that the intervention was successful in improving motivation for handwashing and capability for handwashing...” This conclusion raises questions considering the fact that the study adopted a qualitative approach. You appear emphatic even though you did not seek to determine a causal relationship (which apparently does not fall within the remit of a qualitative study). Furthermore, you have earlier indicated that success was assessed from the perspective of children, who are obviously prone to providing socially desirable responses. Also, I do not find the use of the phrase “In general” to be a scientific language. What precisely do you mean by that? I think your conclusion needs to be re-written.

The Conclusion section has been partially revised on page 26. However, we kept the above statement unchanged because we feel that the Results section provides sufficient information to justify this conclusion.

VERSION 2 – REVIEW

REVIEWER	Ruby Biezen The University of Melbourne
REVIEW RETURNED	21-Sep-2019

GENERAL COMMENTS	Manuscript Number: bmjopen-2019-030947.R1 Title: Factors perceived to facilitate or hinder handwashing among primary students: a qualitative assessment of the Mikono Safi intervention schools in NW Tanzania Thank you for the opportunity to review the revised manuscript. As mentioned previously, this is an extremely important study and I commend the research team for this type of research and by doing
---

	so, hopefully will enable better hygiene and health outcome in Tanzania. I am also pleased to see the improved version and it is reading very well. However, there are occasional grammatical and spelling errors throughout, including one too many spaces before full stops or one too many full stops (ie. Page 7), so make sure the changes are corrected and manuscript read through before submission. I have minor suggestions that the authors might like to consider: Abstract: It would be good to include the study period Key words: Suggest including words like: primary schools, hand hygiene etc Introduction:  • P6: First sentence after 'The Mikono Safi Intervention'- extra close bracket for reference 17 Methods:  • P7: I'm not sure if the word 'parsimoniou' describing the use of COM-B is appropriate. Suggest using another word • P10: spell out numbers less than 10, ie. One male and one female • P11: during coding, were there any disagreement? If so, how did you resolve them? Results:  • What was the duration of the interviews/focus groups? Can you provide mean and range? • P12: third line in the results – 'interviewed twice'- suggest expanding this to enhance clarity, ie. '... whom were interviewed two months after the start of intervention and again 8 months after intervention' • P16: 'chool' should be replaced by 'school' Discussion:  • P21: last sentence in paragraph – Table 3. This is part of your results, shouldn't be in the discussion • P23: You have not mentioned 'education package' previously, suggest either labeling this earlier or changing it to what this referred to previously in the methods section • Please consider including strengths in your study. I'm sure there are some! References:  • Please check your referencing to make sure they are correct. For example, some references are missing page numbers, ie. References 8, 24, 29. 
--	--

REVIEWER	Dr. Emmanuel Appiah-Brempong Kwame Nkrumah University of Science and Technology, Ghana
REVIEW RETURNED	23-Aug-2019

GENERAL COMMENTS	Abstract Should be revised on the basis of the corrections made within the manuscript Introduction "While school-based WASH interventions in low- and middle-income countries have been examined with regards to a wide range of health and educational outcomes, including: diarrhea, respiratory infection, school attendance and health outcomes in the domestic
---

	environment [9-13], these studies have shown that the impact on such outcomes is inconsistent” . You meant to say Inconsistent Methods In your earlier manuscript you did indicate that “...teachers were trained on how to implement handwashing lessons”. You appear to be silent on this in the current version of your manuscript. An elaborate description of this training will be useful for the purpose of replicating this study elsewhere. For example, who conducted the training? What is the qualification of the trainer? What were the objectives of the training sessions? What was the duration of the training? Were all teachers trained? If not, what criteria informed the selection of teachers? And what strategies were used to ensure that training was equally done for all intervention schools? Study Design, Sample and Data Collection Methods [ ] Were the urban and rural schools distributed evenly among the intervention and control schools? Otherwise you could be encouraging selection bias which will ultimately affect the validity of your data adversely. “We also assessed students’ handwashing skills: at the end of each FGD, the field researcher walked with the participating students to the handwashing stations for students to demonstrate how they normally washed hands. The field researcher observed whether students washed both hands with soap.” [ ] What tool was used by the field observer? For eg. was is a checklist? Note that the fact that students are observed washing both hands with soap does not necessarily constitute proper handwashing. There are specific steps that ought to be followed [ ] What was the numerical size of each FGD session? If there were different sizes, then we need to know at least the range. Results “Students understood the link between hand hygiene and infections. They also knew the key times they were required to wash hands” All students? Or some students? How did you assess understanding? Discussion “Our education package also had an impact on the emotional drivers of hygiene behaviours.” [ ] Can you support this with evidence from your results section? Note that you assessed impact from the perspective of the children and teachers. “Data suggest that we were successful in triggering emotional responses in children through our information campaigns”. [ ] Which precise data from your study? and in what way does it suggest this? and from whose perspective?
--	--

	“Our data suggests that while the Mikono Safi intervention was successful in changing capability and motivation”. [ ] Which precise data and in what way does it suggest this?
--	---

VERSION 2 – AUTHOR RESPONSE

Reviewer: 3

Abstract : Should be revised on the basis of the corrections made within the manuscript This has been reviewed and reflects changes made on the other sections of the manuscript

Introduction “While school-based WASH interventions in low- and middle-income countries have been examined with regards to a wide range of health and educational outcomes, including: diarrhea, respiratory infection, school attendance and health outcomes in the domestic environment [9-13], these studies have shown that the impact on such outcomes is inconsistent” . You meant to say Inconsistent
The spelling error has been corrected on page 5 on the manuscript

Methods: In your earlier manuscript you did indicate that “...teachers were trained on how to implement handwashing lessons”. You appear to be silent on this in the current version of your manuscript. An elaborate description of this training will be useful for the purpose of replicating this study elsewhere. For example, who conducted the training? What is the qualification of the trainer? What were the objectives of the training sessions? What was the duration of the training? Were all teachers trained? If not, what criteria informed the selection of teachers? And what strategies were used to ensure that training was equally done for all intervention schools?

After the 1st revisions, we realized that this particular material would fit better in upcoming paper that will describe the intervention design and methods. This manuscript is currently in review. We have added, page on the manuscript a brief information as follows: Prior to implementation in each school, the study team completed a half-day training at each participating study school with 8 to 10 teachers to train teachers on education and messaging activities, determine the schedule for school-based activities, and develop school-specific maintenance plans for handwashing facilities and materials. Full details on the Mikono Safi intervention are described in a forthcoming publication (Makata et al., in review).

Study Design, Sample and Data Collection Methods Were the urban and rural schools distributed evenly among the intervention and control schools? Otherwise you could be encouraging selection bias which will ultimately affect the validity of your data adversely.

The Rural and urban schools were evenly distributed in the control and the intervention arms this information has been added, page 8 on the manuscript. More details on this will be provided on the paper reporting the Mikono Safi trial results expected early 2020

“We also assessed students’ handwashing skills: at the end of each FGD, the field researcher walked with the participating students to the handwashing stations for students to demonstrate how they normally washed hands. The field researcher observed whether students washed both hands with soap.”.

What tool was used by the field observer? For e.g. was is a checklist? Note that the fact that students are observed washing both hands with soap does not necessarily constitute proper handwashing. There are specific steps that ought to be followed

We have provided the following response on page 9 of the manuscript:

Hand washing education sessions had a practical component that took students through 5 hand washing steps which included: i) wetting hands, ii) putting soaps, iii) scrubbing the palms together, iv) scrubbing between the fingers and nails while counting up to 10, and v) rinsing hands with running

water. Handwashing skills for each student were assessed during observation by noting how many of the five recommended steps of handwashing were completed.

What was the numerical size of each FGD session? If there were different sizes, then we need to know at least the range.

The number of FGD participants ranged from 7 to 10 students [see Page 9 of the manuscript]

Results: "Students understood the link between hand hygiene and infections. They also knew the key times they were required to wash hands"

All students? Or some students? How did you assess understanding?

This was qualitative assessment of the Mikono Safi trial from the perspective of teachers and students using a small, purposively selected sample of teachers and students in four out of 8 intervention schools. We are therefore cautious about making generalizing statements about the results of this study. We have therefore revised the paragraph, pages 15 & 16 of the manuscript to read, "During the interviews and focus group discussions we asked students the following question: What happens if you don't wash your hands? Students could explain the link between hand hygiene and infections. The quote below is an example common stories throughout the students' interviews:

If I do not wash my hands, I get diseases especially worms. For example if I eat even a fruit without washing my hands that means I am eating dirt and I will get worms and stomach ache. So whenever you eat without washing hands you get worms (FPI younger girl, school, 3)"

Discussion "Our education package also had an impact on the emotional drivers of hygiene behaviours." Can you support this with evidence from your results section? Note that you assessed impact from the perspective of the children and teachers.

We appreciate the comment and we acknowledge that we are not able to assess impact using the current data because this was a qualitative study reporting students and teachers perceptions about the intervention. We have revised this section to read: "The qualitative data from students and teachers suggest that the Mikono Safi intervention was associated with the targeted emotional drivers of nurture, fear and disgust. In the student data there were rich stories of children describing Muta's poor hygiene and repulsive and disgusting. Students noted that they often washed hands to avoid being labeled "Muta" by their colleagues."

"Data suggest that we were successful in triggering emotional responses in children through our information campaigns". Which precise data from your study? and in what way does it suggest this? And from whose perspective?

The paragraph has been rephrased as follows: Our findings suggest that nurture motive was particularly salient to students. Student spoke fondly of the fact that Koku (the female student character appearing in the education materials) helped her friend out; and both teachers and students discussed the ways the school had provided supportive environment for hand washing.

"Our data suggests that while the Mikono Safi intervention was successful in changing capability and motivation". Which precise data and in what way does it suggest this? This has been rephrase as indicated in 8 and 9 above

Reviewer: 2

Abstract: It would be good to include the study period

This has been included page 1 on the manuscript

Key words: Suggest including words like: primary schools, hand hygiene etc The suggested key words have been included on page 2

Introduction: P6: First sentence after 'The Mikono Safi Intervention' - extra close bracket for reference 17

The extra bracket has been deleted

Methods: P7: I'm not sure if the word 'parsimonious' describing the use of COM-B is appropriate. Suggest using another word

The word parsimonious on page 7 and the sentence has been rephrase to read: COM-B, relative to other behaviour change theories and models, provides an intuitive and flexible approach to understanding behaviours that is applicable across a number of behaviours and contexts

P10: spell out numbers less than 10, ie. One male and one female This has been revised

P11: during coding, were there any disagreement? If so, how did you resolve them?

We have included a sentence to further describe this on page 12 which reads as follows: There were no significant disagreements about how specific codes mapped back to the COM-B Framework. However, there were discussions about codes that conflated one or more COM-B determinants. These codes were then revised into sub-codes that better mapped to an individual COM-B determinant

Results: What was the duration of the interviews/focus groups? Can you provide mean and range?

This has been added on page 13 as follows: On average a student FGD/FPI lasted for 1 hour 40 minutes while teacher's interview was about 40 minutes

P12: third line in the results – 'interviewed twice'- suggest expanding this to enhance clarity, ie. '... whom were interviewed two months after the start of intervention and again 8 months after intervention'

This has been revised to read: In addition, in-depth interviews were held with 16 teachers whom we interviewed two months after the start of intervention and again 8 months after intervention had been rolled out

P16: 'chool' should be replaced by 'school'
This has been revised

Discussion: P21: last sentence in paragraph – Table 3. This is part of your results, shouldn't be in the discussion

The table has been move to the results section, pages 22-23 on the manuscript

P23: You have not mentioned 'education package' previously, suggest either labeling this earlier or changing it to what this referred to previously in the methods section This has been revised on page 13 and the phrase education package has been deleted

Please consider including strengths in your study. I'm sure there are some!

The strength of the study has been stated under Strength and limitation section, page 3 and 27 on the manuscript

References: Please check your referencing to make sure they are correct. For example, some references are missing page numbers, ie. References 8, 24, 29. We appreciate the comments, the references have been revised

VERSION 3 – REVIEW

REVIEWER	Emmanuel Appiah-Brempong KNUST, Ghana
REVIEW RETURNED	03-Nov-2019
GENERAL COMMENTS	Authors need to be more transparent and elaborate on the methods section